# The Effect of Mulberry Silage Supplementation on the Carcass Fatness and Long-Chain Fatty Acid Composition of Growing Lambs Compared with Traditional Corn Silage

**DOI:** 10.3390/foods13172739

**Published:** 2024-08-29

**Authors:** Yang Cao, Xiaoou Zhao, Kaizhi Zheng, Jianliang Wu, Zhiqiang Lv, Xin Huang, Yongqing Jiang, Wenwen Fang, Yang Cao, Junfang Jiang

**Affiliations:** 1Institute of Animal Husbandry and Veterinary, Zhejiang Academy of Agricultural Sciences, Hangzhou 310021, China; cyang0508@163.com (Y.C.);; 2Institute of Animal Husbandry and Veterinary, Jilin Academy of Agricultural Sciences, Changchun 136100, China

**Keywords:** lamb, mulberry, carcass fatness, long-chain fatty acid

## Abstract

Lamb meat has become very popular with consumers in recent years due to its nutritional benefits. As a lean red meat, lamb is an important natural source of polyunsaturated and saturated fatty acids, which can be modified by adjustments in livestock feed. This study used proteomic and metabolic analyses to compare a basal ration supplemented with either mulberry silage or corn silage. Supplementation with mulberry silage led to a reduction in subcutaneous carcass fatness compared with corn silage. Additionally, changes in the proteome associated with fatty acid metabolism and oxidation resulted in decreased levels of saturated and trans fatty acids, while significantly increasing the levels of α-linolenic acid (ALA) and oleic acid and reducing linoleic acid content.

## 1. Introduction

As the demands for quality of life gradually increase, people need not only meat intake but also meat with better taste and higher nutritional value. Lamb meat contains a variety of polyunsaturated fatty acids which not only affect its taste but are also essential nutrients for humans [1]. Red meat and seafood are major sources of long-chain fatty acids in the diets of adults [2]. Lean red meat is an important natural dietary source of long-chain omega-3 polyunsaturated fatty acids (Ω-3 LCPUFAs) [3]. Lamb fat comprises saturated or monounsaturated fatty acids. Polyunsaturated fatty acids only comprise a minor proportion of lamb fat. Saturated fats are more stable and less prone to oxidation [4]. Excessive consumption of saturated fatty acids can increase blood cholesterol levels (LDL) [5,6] and can also increase the risk of cardiovascular and cerebrovascular disease [7,8]. To maintain good health, it is therefore recommended to minimize the intake of saturated fatty acids.

The mulberry is from the Moraceae family of trees that grows natively in various parts of China. It is highly adaptable, has a rapid growth rate, and has a long lifespan [9]. Therefore, it is widely planted throughout China and is suited to the weather of the country. Mulberry leaf extract contains flavonoids, alkaloids, polysaccharides, proteins, and other active ingredients. It also has pharmacological importance in such areas as improving blood glucose [10], blood lipid [11], and antioxidation levels [12]. To improve the nutritional value of mulberry leaves, fermentation treatment could have an important effect. The hard-to-digest macromolecules can be converted into digestible small molecules such as small peptides and amino acids; the content of anti-nutritional factors such as phytic acid can be significantly reduced; the digestibility of nutrients can be improved; and the palatability can be enhanced [13]. The addition of fermented mulberry leaves and unfermented mulberry leaves in the feed can also increase the amount of feed intake. However, as compared to unfermented mulberry leaves, fermented mulberry leaves are more beneficial for reducing the energy density of the feed. [14]. The flavonoid values of mulberry leaves have great potential to regulate the microbiome, fermentation process in, and metabolism of ruminants, improving ruminant performance and health, and reducing CH_4_ emissions [15]. Studies also suggest that feeding 10–15% mulberry silage can increase the concentration of polyunsaturated fatty acids in milk content [16]. These studies may indicate that adding mulberries to the feed can increase long-chain fatty acid concentration in animal meat. Hu sheep are a widely raised breed in China. Therefore, our experimental design is based on Hu sheep. We conducted a four-month mulberry silage feeding trial, comparing it with traditional corn silage feed, to study the effect of feeding mulberry silage on the subcutaneous carcass fat and long-chain fatty acid composition of Hu sheep.

## 2. Material and Methods

### 2.1. Animals

This experiment was designed with two treatments (corn silage and mulberry silage); each treatment contained five replicates, and each pen was one replicate. Forty 4-month-old male Hu sheep were randomly divided into two groups under the same conditions. Each group was divided into 5 pens, with 4 individuals in each pen. All of the lambs were fed a diet consisting of concentrated feed (0.2 kg per lamb) and peanut vines (0.06 kg per lamb) as basal feed, with corn silage (0.5 kg per lamb as a control) or mulberry silage (0.5 kg per lamb) added as supplementary feed in each group (Table 1). The supplementary feed and basal feed were mixed evenly by weight, with 0.8 kg of concentrated feed, 0.24 kg of peanut seedlings, and 2 kg of supplementary feed per pen every time. The lambs were fed twice a day. After four months of feeding, blood samples of all lambs were taken, and 23 blood-related indicators were tested. After the separation of the serum, four blood lipids, including triglycerides (TGs), total cholesterol (CHO), high-density lipoprotein (HDL), and low-density lipoprotein (LDL), were analyzed. One lamb from each pen was randomly selected for slaughter, meaning that we selected five lambs for each treatment. The slaughter took place on the assembly line of the local commercial abattoir, and the slaughter method followed traditional procedures, including stunning. After slaughtering, the subcutaneous adipose tissues were separated and weighed, and the longissimus thoracis was used to measure meat-related indicators. The adipose tissue was frozen in liquid nitrogen and sent to a biological company for sequencing.

The moisture, protein, fat, and ash content of the longissimus thoracis were measured by the Quality Inspection Center for Agricultural and Processed Product Safety of the Ministry of Agriculture and Rural Affairs of China. The moisture content was determined according to GB 5009.3-2016, the crude ash content was determined according to GB 5009.4-2016, the crude protein content was determined according to GB 5009.5-2016, and the crude fat content was determined according to GB 5009.6-2016 [17].

### 2.2. DIA Proteome and PRM

Proteins were extracted from adipose tissue at 4 °C and their concentrations were determined using the BCA (bicinchoninic acid) method. The sample preparation included denaturation, reduction, and alkylation of the isolated proteins as well as tryptic digestion and peptide purification. Chromatographic separation was performed using the UltiMate 3000 liquid chromatography system (Thermo Fisher Scientific, Waltham, MA, USA) coupled to a timsTOF Pro 2, a mass spectrometer with ion-mobility spectrometry and quadrupole time-of-flight detection. Data independent acquisition (DIA) was processed and analyzed using a Spectronaut 17 (Biognosys, Zurich, AG, Switzerland) with default settings. Trypsin was used as the digestive enzyme, and carbamidomethyl on cysteine was specified as the fixed modification.

The proteins of interest were selected from the proteome sequencing results for targeted testing by parallel reaction monitoring (PRM). Mass spectrometric data acquisition was performed using a Q Active HF mass spectrometer coupled to the UltiMate 3000 RSLCnano system for liquid chromatography–mass spectrometry (LC-MS). The peptide samples dissolved in a loading buffer were inserted via an autosampler and separated on an analytical column (75 μM × 25 cm, C18, 1.9 μm, 120A). The chromatographic gradient was established with two mobile phases: mobile phase A (0.1% formaldehyde and 3% DMSO) and mobile phase B (0.1% formaldehyde, 3% DMSO, and 80% acetonitrile). The flow rate of the mobile phase was kept at 300 nL/min. Data acquisition in the mass spectrometer was set to data-dependent acquisition (DDA) mode, which included a full MS scan (R = 60 K, AGC = 3 × 10^6^, max IT = 25 ms, scan range = 350–1500 *m*/*z*), followed by 20 MS/MS scans (R = 15 K, AGC = 1 × 10^5^, max IT = 50 ms). Higher-energy collisional dissociation (HCD) was applied with a collision energy of 27, and an isolation window of 1.4 Da was used for the quadrupole. The dynamic exclusion was set to 24 s to prevent the repeating of ion selection.

### 2.3. Targeted Detection of Long-Chain Fatty Acids

For preparation, the sample was transferred to a 20 mL glass centrifuge tube and 5 mL of 10% acetylchloromethanol, 1 mL of n-hexane, and 5 μL of internal standard (C19:0, 10 mg/mL) were added. The mixture was allowed to react for 4 h at 95 °C and 250 rpm. Subsequently, 6 mL of 6% potassium carbonate solution was added to the mixture and vortexed for 2 min, followed by centrifugation to separate the n-hexane, which was then removed by rotary evaporation. An additional 400 μL of n-hexane was added to it. After another round of vortexing for 1 min and centrifugation at 12,000 rpm/min for 5 min, the clear supernatant was collected into an injection vial. The prepared samples were then analyzed using a Thermo Trace1300 gas chromatograph connected to an ISQ7000 mass spectrometer (GC-MS) (Thermo Fisher Scientific, MA, USA) to identify and quantify the components.

These data were extracted from GC/MS data using the Thermo Data Software Chromeleon 7.0 and organized into a two-dimensional data matrix based on the NIST 17 database, including retention time, sample, and peak intensity.

### 2.4. Statistical Analysis

Differentially expressed proteins were identified using Student’s t-test and then adjusted for multiple comparisons using the Benjamini and Hochberg (BH) method. The proteins that met the criteria of a fold change >2 or <2 and a Q value < 0.05 were considered significantly differentially expressed.

Protein functions and classification were analyzed using queries in the following databases: Gene Ontology (GO), EuKaryotic Orthologous Groups (KOG)/Cluster of Orthologous Groups of Proteins (COG), and the Kyoto Encyclopedia of Genes and Genomes (KEGG) database.

Data were analyzed using Maxquant software (V2.4.2.0) for initial retrieval from the database. Quantitative analysis of target proteins and peptides was then performed using Skyline software (v17.0), which provides detailed quantitative information important for PRM validation, as follows: Use Skyline software to perform targeted extraction of peptide segments of the target protein in each sample raw file. Select 3 sub ions with the best signal from each peptide segment for quantification. Normalize the intensity which was detected by mass spectrometry to obtain quantitative information of the target peptide segment in different samples. Take Log2 for its quantitative value and compare the differences between the two groups of data [18].

Unless otherwise stated, data were expressed as mean ± SEM. The number of replicates was 3 for proteomics and PRM and 5 for meat quality assessment and GC/MS. Comparing the two sets of data, a t-test was used for statistical analysis.

## 3. Results and Discussion

### 3.1. The Effect of Feeding Mulberry Silage on Carcass Fatness and Meat Quality

When comparing mulberry silage with corn silage as a supplementary feed for lambs (Figure 1A), the blood of the lambs was taken after 4 months of feeding to determine 23 routine blood indicators, including white blood cell count, lymphocyte count, hemoglobin, etc. It was found that, except for the mean corpuscular hemoglobin concentration, the blood indicators showed no significant changes. Although the mean corpuscular hemoglobin concentration showed significant differences, it was within the normal range (Table 2). Four indicators of blood lipids, including high-density lipoprotein (HDL), low-density lipoprotein (LDL), total cholesterol (CHO), and TG, were determined, of which HDL increased significantly, while the other indicators showed no significant differences (Figure 1B). Reverse cholesterol transport is the process of transporting cholesterol back to the liver from extrahepatic tissues. The lipoprotein HDL is a key mediator of reverse cholesterol transport [19]. The increase in HDL may have been caused by a reduction in fat deposition. Weight loss through a low-fat diet is associated with an increase in HDL cholesterol levels, which has a positive effect on cardiovascular and cerebrovascular health [20]. Therefore, in this research, there is a certain correlation between elevated HDL and decreased subcutaneous fat.

Five lambs were randomly selected from each group to be slaughtered and tested for carcass fatness. The result showed that the weight of the subcutaneous fat decreased significantly (Figure 1C) and the intramuscular fat content stayed consistent (Figure 1D) (Table 3). The results showed that the addition of mulberry silage to the diet could reduce the subcutaneous fat content of the lambs and had no effect on the intramuscular fat content and tenderness of the lamb meat. Recent studies have shown that adding mulberry leaf flavonoids to the feed of fattening pigs could reduce the subcutaneous adipose and increase the visceral fat of the pig [21]. The reduction in high-density lipoprotein subclasses increases visceral adipose tissue and reduces peripheral adipose mass [22]. The reduction in carcass fatness and the unchanged intramuscular fat in the group of lambs that were fed mulberry may be due to changes in HDL.

### 3.2. Proteome Analysis of Carcass Fatness in Sheep Fed with Mulberry Silage

Feeding on mulberry silage resulted in a decrease in the fat content of the lambs. To further determine the cause of this phenomenon, proteomic sequencing was performed to determine the difference in adipose tissue between the lambs in the mulberry silage group and the control group. Proteomic sequencing identified 601 differentially expressed proteins, including 419 upregulated and 182 downregulated proteins (Figure 2). These proteins are enriched in disease-related metabolic pathways such as Parkinson’s disease, Alzheimer’s disease, Huntington’s disease, etc. The common differences in these disease-related metabolic pathways depend mainly on the generation of reactive oxygen species (ROS) in the mitochondria. Significant changes take place in protein expression levels related to ROS production, such as COX6BI [23], COX4I1 [24], COX6A1 [25], COX5B [26], NDUFS4 [27], NDUFS2 [28], NDUFS3 [29], and NDUFS1 [30] (Figure 3). Numerous studies have shown a significant link between ROS and neurodegenerative diseases such as Alzheimer’s disease [31], Parkinson’s disease [32], Huntington’s disease [33], Machado–Joseph disease [34], and senescence [35]. Researchers have developed many neuroprotective therapies to combat ROS that protect neuronal cells and block neurodegenerative diseases. Previous research has identified biomarkers of oxidative damage associated with senescence and neurodegenerative diseases. These biomarkers include damaged biomolecules such as lipids, proteins, and DNA [36]. Some studies suggest that an increase in ROS levels can affect the physiological state of red blood cells and damage hemoglobin in red blood cells [37,38]. This explains the changes in hemoglobin levels of blood cells in the previous section’s results, as the changes in ROS levels alter the concentration of hemoglobin in red blood cells. Research indicates that ROS contribute to various oxidative processes, including lipid peroxidation to malondialdehyde (MDA), the formation of protein carbonyls, and oxidation of guanine to 8-hydroxydeoxyguanosine in DNA. These processes can adversely affect lipids [39]. Consequently, the reduction in carcass fatness observed in this study can also be attributed to ROS-mediated lipid oxidation.

In addition, differential proteins were also enriched in metabolic pathways such as oxidative physiology; thermogenesis; valine, leucine, and isoleucine catabolism; fatty acid metabolism; the citrate cycle; fatty acid degradation; fatty acid elongation; and many others (Figure 4). Using PRM to validate the partial proteomic data in the above metabolic pathways showed a trend consistent with the proteomic data (Figure 5). The changes in the fatty acid-related metabolic pathways caught our attention. Feeding mulberry silage resulted in changes in proteins related to fatty acid oxidation and metabolism such as FASN, OXSM, ELOVL6, ACAA1, ACOX1, and the ACSL family. Studies have shown that the addition of mulberry leaves to a high-fat diet in rats can induce fatty acid oxidation, inhibit fat formation, and prevent oxidative stress. The contents of mulberry can induce genetic changes in fatty acid oxidation pathways such as CPT1 and ACAA1 while decreasing the synthesis of fatty acid proteins such as ELOVL2 and OXSM and affecting the TCA cycle [40]. Some recent studies suggest that mulberry leaves alleviate inflammation and carcass fatness by promoting AMPK signaling while supporting lipid metabolism and fatty acid oxidation. The inhibitory effect of mulberry leaves on obesity in mice is due to the downregulation of TNF-α, PPARD, and PPARG and the upregulation of FAAH and HSD1B. Mulberry leaves can regulate lipid metabolism and catabolism, fatty acid metabolism and biosynthesis, and inflammatory responses, thereby reducing obesity [41]. Research has shown that adding mulberry leaf powder to the feed of fattening pigs can affect the expression levels of lipid metabolism genes HSL, ACC α, LPL, and PPAR γ in muscle tissue, as well as the expression of mitochondrial uncoupling proteins [42]. Adding mulberry leaves to chicken feed can alleviate hepatic steatosis and improve lipid metabolism by downregulating circACACA [43]. These studies indicate that adding mulberry to feed can affect the fatty acid metabolism and oxidation levels in animals through these lipid metabolism-related genes and proteins, thereby affecting fat production and fatty acid composition.

### 3.3. Targeted Metabolomics Detection of Long-Chain Fatty Acids in Adipose Tissue

By detecting changes in long-chain fatty acids using targeted metabolomics, a total of 46 long-chain fatty acids were detected, including 11 differentially expressed long-chain fatty acids, with 2 upregulated and 9 downregulated. The results showed significant differences in long-chain fatty acids such as methyl laurate, methyl tricosanoate, methyl myristate, methyl myristoleate, methyl palmite-laidate, methyl linoleate, methyl alpha linolenate, methyl oleate, methyl arachidonate, methyl 11-14-17 eicosatrienoate, methyl docosahexaenoate, and others (Figure 6). The content of most saturated fatty acids in the results, such as methyl undecanoate, methyl laurate, methyl tridecanoate, methyl myristate, etc., decreased, while the content of neurotransmitters increased. The content of trans fatty acids, such as methyl linolelaidate, methyl transvaccenate, methyl palmite-laidate, methyl linolelaidate, methyl transvaccenate, and methyl trans 11-eicosenoate, decreased. The effects of saturated fatty acids on human health have always been controversial [44,45,46], but it can be concluded that saturated and trans fatty acids are not suitable for people who want to lose weight [4,47]. The content of the majority of saturated fatty acids and trans fatty acids in the adipose tissue of lambs that were fed mulberry silage decreased. There were also changes in monounsaturated fatty acids and polyunsaturated fatty acids. Studies have shown that mulberry leaves alter the fatty acid composition of adipose tissue in obese mice that have a high fat intake, significantly increasing the ratio of polyunsaturated fatty acids to saturated fatty acids [48]. The proportion of monounsaturated fatty acids in the adipose tissue of obese mice fed mulberry leaves increased, while the proportion of polyunsaturated fatty acids decreased [49]. Using mulberry leaf extract as a supplementary diet, the total content of unsaturated fatty acids and polyunsaturated fatty acids in broiler chickens increased, while the content of saturated fatty acids decreased [50]. This indicates that mulberry can regulate the fatty acid composition in animals.

A combined analysis of differential proteins enriched in the fatty acid metabolism pathway and differential metabolites detected by targeted long-chain fatty acids revealed that many differential proteins were correlated with arachidonic acid, linoleate, and ALA. Proteins such as ELOVL6, ACAA2, ACOT4, and ACACA collectively regulated methyl linoleate and methyl arachidonate (Pearson coefficient greater than 0.9, *p* < 0.01, Figure 7). Many studies have shown that ELOVL6, ACAA2, ACOT4, and ACACA are associated with adipogenesis and fatty acid metabolism [51,52,53,54,55]. The ELOVL6 gene plays an important role in the synthesis of long-chain saturated fatty acids and monounsaturated fatty acids [56]. Knocking down ELOVL6 in 3T3-L1 cells resulted in a significant increase in palmitic acid and a significant decrease in oleic acid content [57]. Research has shown that feeding goats with mammary glands with highly concentrated feed reduces the expression levels of ACACA as well as LPL involved in lipid metabolism, leading to changes in the content of linoleic acid in milk [58]. Inhibiting ACACA activity does not reduce the rate of de novo fat generation, but it reduces the synthesis of long-chain fatty acids [59]. To understand the regulatory relationship between these proteins and arachidonic acid, linoleate is worthy of further research.

The experimental results showed that, in addition to a decrease in the content of saturated fatty acids and trans fatty acids, the unsaturated fatty acids such as α-linolenic acid (ALA) and oleic acid increased significantly by 39% and 37%, respectively, and these are important for human health. Linoleic acid, on the other hand, decreased significantly by 43%. The rumen hydrogenation of ruminants can affect the status of oleic, linolic, and alpha linolenic acids [60]. Oleic acid can regulate blood lipid levels, lower cholesterol levels [61], and effectively reduce the incidence of hypercholesterolemia and cardiovascular disease and the risk of coronary heart disease [62]. Similarly, most people need to increase their dietary intake of Ω-3 LCPUFAs to reduce the risk of chronic disease [63]. ALA is an essential Ω-3 LCPUFA for the human body, but α-linoleic acid cannot be synthesized by the human body and can only be obtained from the diet [64]. It is also a synthetic substrate which can be metabolized in the body and converted to EPA and DHA [65]. The conversion of dietary ALA to DHA is sufficient to supply the brain [66]. ALA inhibits the metastasis and proliferation of osteosarcoma cells by downregulating the expression of FASN [67], and dietary intake of ALA is associated with a lower risk of death from cardiovascular disease and coronary heart disease [68]. The recommended intake of ALA is 1.1–2.2 g/day; specifically, for men it is recommended that they consume 0.6–2.2 g/day and for women 0.5–2.1 g/day. This means that the dietary intake of ALA is insufficient in some people [69]. The intake of Ω-6-LCPUFAs and Ω-3-LCPUFAs is related to metabolic health. Inadequate intake of long-chain Ω-3 fatty acids can lead to metabolic disorders, and the ratio of Ω-6 LCPUFA to Ω-3 LCPUFA intake is important for human health [70]. Supplementing the diet with Ω-3 LCPUFAs is more helpful for the treatment of obese individuals, especially women [71]. Low intake of Ω-3 LCPUFAs has been found in obese individuals, and the ratio of Ω-6 LCPUFAs to Ω-3 LCPUFAs correlates with the production of pro-inflammatory factors that can cause an inflammatory response that turns metabolically healthy obese individuals into metabolically unhealthy obese people [72]. In recent years, people’s dietary habits of consuming vegetable oil and deep-frying have led to excessive intake of linoleic acid and minimal intake of ALA, resulting in an unequal ratio of linoleic acid to ALA intake. In the present study, the content of ALA in the fat of lambs fed mulberry leaves increased, while the content of linoleic acid decreased [73]. This suggested that the intake ratio of ALA to linoleic acid was therefore more in line with the requirements of a healthy diet.

## 4. Conclusions

The inclusion of mulberry silage in the feed of growing lambs can reduce carcass fatness, specifically lowering the content of saturated and trans fatty acids in the adipose tissue, without affecting intramuscular fat content and meat tenderness. The fatty acid composition in the adipose tissue of lambs fed with mulberry silage changes, with a decrease in the content of saturated and trans fatty acids. Additionally, mulberry silage significantly increases the content of alpha-linolenic acid (ALA) and oleic acid, while reducing the linoleic acid content. This effect is facilitated through pathways related to fatty acid metabolism and oxidation. As a result, the lamb meat obtained from such feeding processes is of higher quality and can better meet the nutritional needs of consumers.

## Figures and Tables

**Figure 1 foods-13-02739-f001:**
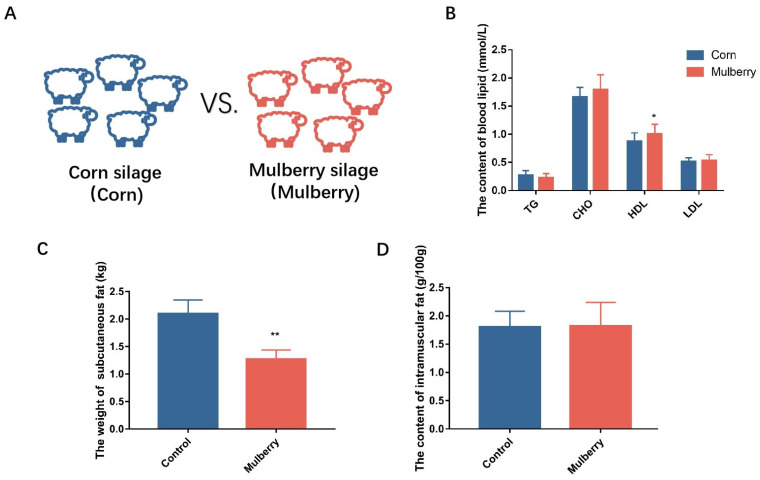
Mulberry and corn silage feeding experiment. (**A**) Feeding schematic diagram; (**B**) blood lipid content, high-density lipoprotein (HDL), low-density lipoprotein (LDL), and total cholesterol (CHO); (**C**) the weight of subcutaneous fat; (**D**) the content of intramuscular fat content. * *p* < 0.05, ** *p* < 0.01.

**Figure 2 foods-13-02739-f002:**
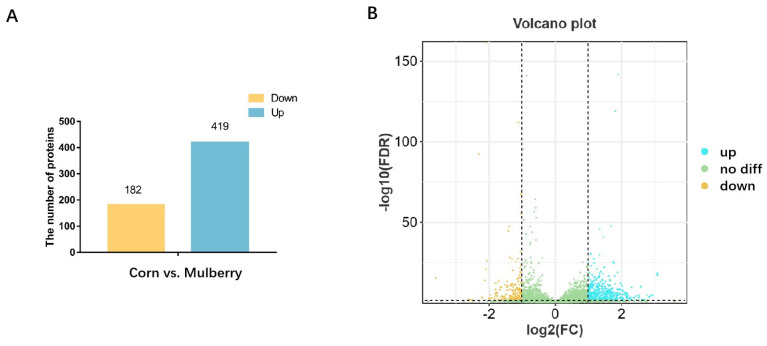
Bar chart (**A**) and volcanic map (**B**) of differential proteins.

**Figure 3 foods-13-02739-f003:**
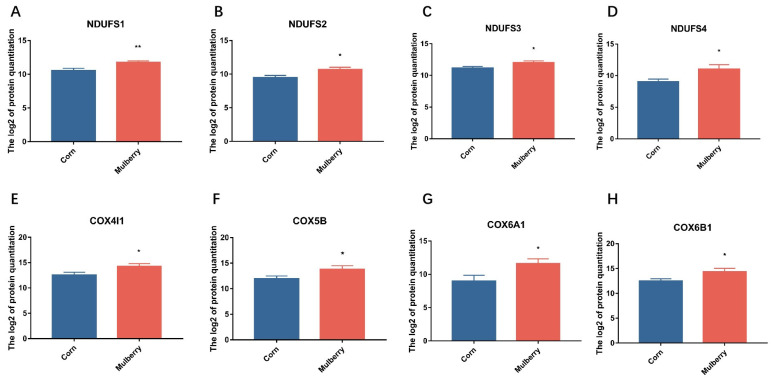
Differential proteins related to ROS in the proteome between corn silage and mulberry silage. (**A**–**H**): Protein expression levels of NDUFS1, NDUFS2, NDUFS3, NDUFS4, COX4L1, COX5B, COX6A1 and COX6B1. * *p* < 0.05; ** *p* < 0.01.

**Figure 4 foods-13-02739-f004:**
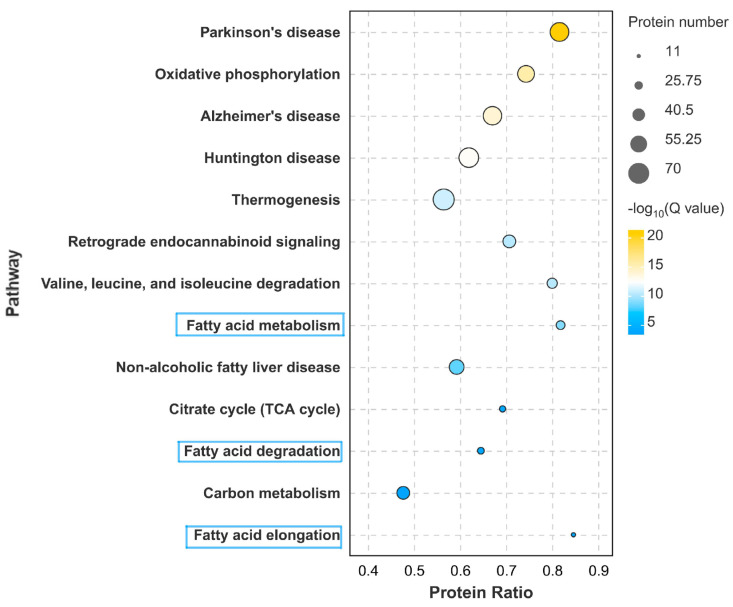
KEGG analysis of differential proteins in proteome.

**Figure 5 foods-13-02739-f005:**
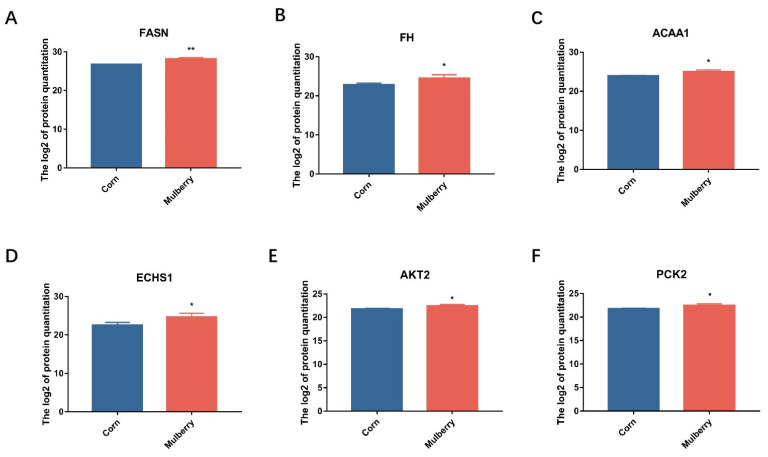
PRM validation of differential proteins between corn silage and mulberry silage. (**A**–**F**): Protein expression levels of FASN, FH, ACAA1, ECHS1, AKT2, and PCK2. * *p* < 0.05; ** *p* < 0.01.

**Figure 6 foods-13-02739-f006:**
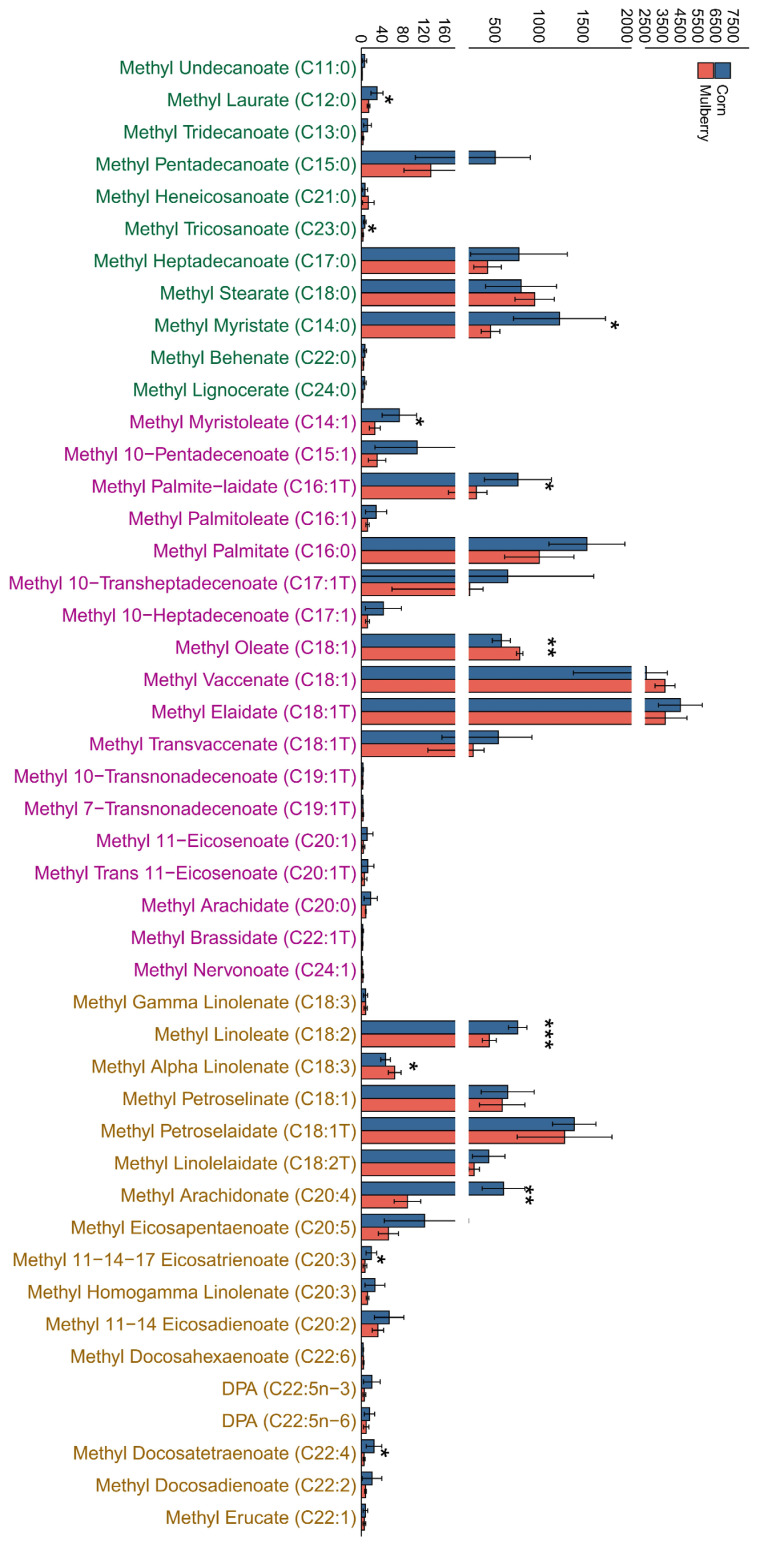
The content of long-chain fatty acids in adipose tissue of lambs fed corn silage and mulberry silage. * *p* < 0.05; ** *p* < 0.01; *** *p* < 0.001.

**Figure 7 foods-13-02739-f007:**
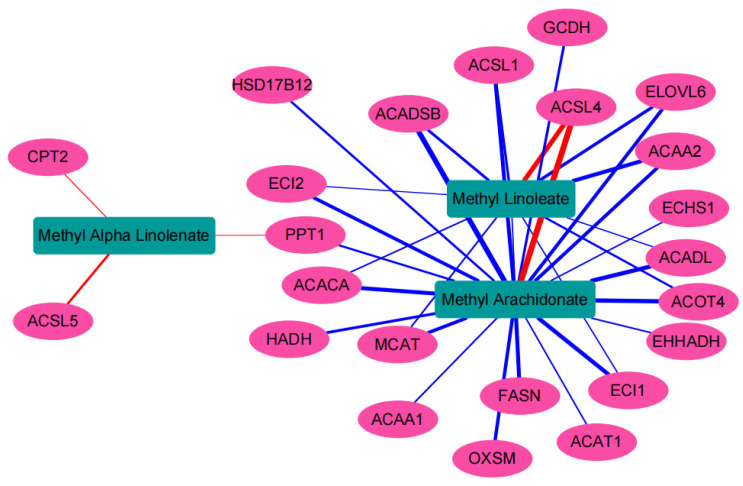
Correlation analysis between linoleate, alpha linolenate, arachidonate, and differential proteins in fatty acid metabolism pathways. Red lines indicate positive correlation, blue lines indicate negative correlation, and the thickness of lines represent level of correlation.

**Table 1 foods-13-02739-t001:** The ration components (%).

Items/Group	Control	Mulberry
concentrated feed	21.3%	21.3%
peanut vines	25.5%	25.5%
corn silage	53.2%	0
mulberry silage	0	53.2%

**Table 2 foods-13-02739-t002:** The blood indicators of experimental lambs.

Group	Corn	Mulberry
White blood cells	8.41 ± 5.18	11.94 ± 2.63
Neutrophils	1.86 ± 1.62	2.78 ± 0.80
Lymphocytes	4.79 ± 2.60	6.48 ± 2.36
Monocytes	0.98 ± 0.70	1.63 ± 0.329
Eosinophils	0.70 ± 0.50	0.95 ± 0.29
Basophils	0.07 ± 0.05	0.09 ± 0.029
Neutrophils (%)	18.42 ± 8.19	23.76 ± 6.99
Lymphocytes (%)	60.44 ± 9.35	53.06 ± 11.25
Monocytes (%)	12.06 ± 3.63	14.10 ± 3.43
Eosinophils (%)	8.28 ± 2.45	8.34 ± 3.18
Basophils (%)	0.80 ± 0.10	0.74 ± 0.089
Red blood cells	10.44 ± 1.70	11.46 ± 1.20
Hemoglobin	99.40 ± 16.29	114.80 ± 9.68
Erythrocytes	32.80 ± 5.36	36.24 ± 3.19
Average red blood cell volume	31.46 ± 1.52	31.70 ± 1.60
Average hemoglobin content of red blood cells	9.54 ± 0.35	10.04 ± 0.52
Mean corpuscular hemoglobin concentration	303.20 ± 10.18	317.00 ± 7.62 *
Coefficient of variation of red blood cell distribution width	17.90 ± 0.83	18.76 ± 0.65
Standard deviation of red blood cell distribution width	21.00 ± 1.85	22.20 ± 1.31
Platelets	202.20 ± 138.58	219.20 ± 72.76
Average platelet volume	4.60 ± 0.31	4.60 ± 0.32
Platelet distribution width	15.14 ± 0.52	14.92 ± 0.36
Platelet hematocrit	0.10 ± 0.07	0.10 ± 0.04

Note: * indicates significant differences between groups (*p* < 0.05) and *p* < 0.05 is considered statistically significant.

**Table 3 foods-13-02739-t003:** The fat data of experimental lambs.

Group	Corn	Mulberry
Subcutaneous fat (kg)	2.10 ± 0.15	1.27 ± 0.10 **
Subcutaneous fat rate (%)	8.81 ± 0.34	5.38 ± 0.32 **
The content of intramuscular fat (g/100 g)	1.80 ± 0.13	1.82 ± 0.19

Note: ** indicates extremely significant differences between groups (*p* < 0.01), and *p* < 0.01 is considered extremely significant.

## Data Availability

The original contributions presented in the study are included in the article, further inquiries can be directed to the corresponding author.

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
