# Peer review of "The Effect of Mulberry Silage Supplementation on the Carcass Fatness and Long-Chain Fatty Acid Composition of Growing Lambs Compared with Traditional Corn Silage"

_foods, 2024, doi:10.3390/foods13172739_

Round 1

Reviewer 1 Report

Comments and Suggestions for Authors

I congratulate the authors on the topic, its scientific interest, as well as the search for endogenous resources, mulberry trees, as a nutritional source for animal feed, with a view to the quality of the final product and its impact on the human diet and new trends.

The introduction is brief, very objective and concise, based on two main points - the quality of lamb meat and the possible properties of blackberries. However, many of the considerations mentioned are not bibliographically supported (lines 29-35) and some could be supported by more than one article, given their relevance to the work and implication in the conclusions (particularly those referring to mulberry). On the other hand, I think this is the place to present other studies that have already used mulberry silage (which are covered in the results and discussion), obviously as an introduction to the topic (lines 226-233) of its benefits in animal feed, in order to contextualise the research. 

With regard to materials and methods, a more detailed explanation is essential. As for the animals, a brief reference to the facilities and management during the study months, as well as the characteristics of the animal breed chosen. The indicators should be referenced (line or supplement), and the process of slaughtering, collecting and processing the samples should be more detailed. Considering that the results are directly influenced by the physicochemical analysis of the meat, it is not enough to refer to the measurement (methods, equipment, ....). As for DIA Proteome and PRM, the description is correct but the introduction of the statistical methodology of the process could be referred to in point 2.4 Statistical Analysis (line 83-89; 104-107), as it would make it easier to understand the description of both techniques and methodologies. As for the Statistical analysis, which is simple, it would be relevant which statistical package and confidence intervals were used. 

In 3.1 - the results of the differences in blood routine indicators are not clearly explained (mean haemoglobin concentration of red blood cells, mean corpuscular haemoglobin concentration), and the support for the HDL results (137-141) needs more scientific robustness (supporting articles). Furthermore, it has now been realised that the meat quality comes from a smaller sample (not mentioned in the material and methods), which changes the experimental design. Even so, the presentation of the results is unclear, particularly when it comes to the role of mulberry silage in intramuscular fat content (lines 149 and 153), which requires more depth, clarity and justification.

3.2 - The significant contribution of the group fed mulberry silage is unclear. The different metabolic levels and biomarkers are mentioned in general, but the specific results are not shown. Part of the discussion is based on a study (19) (20), which is clearly lacking in terms of the relevance of the information it is intended to convey.

3.3. - The differences between the groups are of different significance and their discussion necessarily deserves to be different because, given the sample, they have a diverse scientific impact (205-208; 213). There are also some statements that deserve to be supported or better clarified (218-221; 223-226; 247-250; 259; 262-263), some of them by just one author and of not very recent date.

Conclusion - Assertive conclusion, but based on the results and discussion, it falls short. The conclusion should have been even more expressive in terms of the specific benefits of using mulberry silage. It is also suggested that further bibliographical research be carried out, admitting that there is some scarcity in terms of silage but not in terms of the different components (proteins and fatty acids) and the effects on health. I would like to congratulate the authors again on the quality and relevance of their work, as well as on the quality of their scientific writing.

Comments on the Quality of English Language

I would like to congratulate the authors on the quality of their scientific writing and proposed small changes.

Reviewer 2 Report

Comments and Suggestions for Authors
The research topics undertaken in this article are interesting. A particularly interesting part concerns less fatness, which the authors attribute to ROS-mediated lipid oxidation. The work of scientists and analyzes has been extensive and implemented.  
"analyzes" (line 12) should be "analyses"
"Long-chain fatty acid" (linia 19) should be "Long-Chain Fatty Acid".

Reviewer 3 Report

Comments and Suggestions for Authors

Effect of Mulberry silage on fat accumulation and long chain fatty acid content in sheep

by

Cao etal

A more appropriate title would be

The effect of mulberry or corn silage supplementation on carcass fatness and fatty acid composition of growing lambs.

General Comments

Overall the abstract makes no attempt to quantify the magnitude of the C18 fatty acid and therefore it is difficult to gauge their impact of any changes on human heath.

For the duration of the experiment (4 to 8 months of age) the lambs would be functioning ruminants (as opposed to milk fed lambs which could use the oesophageal groove to deliver feed components directly to the abomasum). Therefore in this experiment the effect of diet on fat composition would depend upon the amount of dietary material that would escape hydrogenation in the rumen. The effect of diet would be influenced by a number of factors including the amount of mulberry or corn silage fed? Effectively hydrogenation in the rumen would convert the unsaturated C18 fatty acids to saturated oleic acids.

The hypothesis as stated by the authors is inadequate. Need to clearly state what the hypothesis is. Surely it is testing whether long chain fatty acid content is different in fat and lean tissue of lambs feed a mulberry or corn silage supplement prior to slaughter???

Were sheep kept in the same or different pens? If the lambs from the corn or mulberry treatments were fed in separate pens this becomes the experimental unit and therefore the experiment only has 2 degrees of freedom. The 20 animals in each pen represent sampling error, not experimental degrees of freedom. If this is the case the experiment in unanalysable and must be rejected.

The details on the M&M fall short of what is acceptable and needs rewriting.

No data on feed intake of the silage supplementary feeds.

There are no results on the amount of the mulberry and corn silage supplements offered to the lambs, the effect on growth rate, hot carcass weight and subcutaneous fat %. Why did the experiment start with 20 lambs per treatment and only slaughter 5??

Specific Comments

Line 10 - the sheep were 4 months old therefore they need to be classed as young sheep or lamb

Line 11 - ...sheep meat is an important source of long chain ....

Line 12 - This study investigated the effects of supplementing a basal ration with mulberry, or corn silage. Supplementation with mulberry silage lead to a reduction in subcutaneous fat ....

Line 20 - 8 month old lambs would certainly not be classified as mutton?? Because the study does not measure initial fatness no reference can be made to fat accumulation??

Line 24 - this study refers to lamb not mutton

Line 27 - Lean meat, intermuscular and subcutaneous fat is an important dietary source of long chain fatty acids and its content can be influenced by changing the composition of livestock feed.

As I point out in general comments this sentence tends to ignore the effect of hydrogenation in the rumen

Line 29 - the authors appear fixated on referring to the carcasses as mutton??

Line 40 - .....such as “improving” blood glucose level..... Does “improving” mean increasing or decreasing blood glucose level??

Line 50 - the hypothesis needs rewriting.

Material and Methods

Line 55 - the details in the 1st paragraph fall well short of what is acceptable. How much of the basal ration was fed. How was the silage (both corn and mulberry) produced? Was fermentation promoted by wrapping the corn or silage in plastic or was it buried to exclude oxygen. How was the supplement fed to the animals, individually, or in a communal trough? What was the average intake of the basal ration and/or the supplement? Did all animals eat the supplement?

How long were the animals transported from the research station to the abattoir? How was subcutaneous fat defined in this experiment?? Did the same boner excise subcutaneous fat from all the carcasses. What portion of the M longissimus dorsi (LD) was excised from the carcasses? Was the LD trimmed of all connective tissue and intermuscular fat.

Line 70 - need to quote the animal ethics approval number, not just the the Committee.

Line 74 - what does the acronym BCA stand for.

Line 83 - The authors obviously used number of animals as degrees of freedom. If animals were kept in different pens this was not correct.

Line 123 - I assume that 3 replicates for proteomics and the 5 for meat quality were meaned for each animal.

Line 125 - I do not know what this sentence means?

Line 144 - Figure 1 - what was the purpose of the 5 sheep in the 2 treatments in Figure 1??

C - the units for the vertical axis are missing. Why is this not in Supplementary Table 2.

Line 146 - This is the first the authors have mentioned that a selection of 5 animals from each treatment were slaughtered for meat quality. Previously the M&Ms inferred that 20 in each treatment were slaughtered for all traits. Certainly not mentioned in the M&Ms.

Line 150 - where were the methods for shear slice, pressure loss and cooking loss mentioned in the M&M???

Line 155 - this stduy did not measure fat accumulation, rather it measured subcutaneous fat weight at slaughter after a 4 month feed trial.

Section 3.2 - I doubt whether the authors can extrapolate from human diseases to proteins that were up or down regulated in ruminant lambs??

Line 177 - Figure 2 - The axes on graph B are too small to be legible?

Line 200 - what is the conversion to log2. It is not referenced in the Stats method??

Line 218 - Growing lambs are not a suitable model to extrapolate to human conditions

Line 235 - Figure 5 requires a massive input before it is acceptable. Firstly the text is too small, the colour selected in too light and rather than full names for proteins it should include the formulae for the proteins? The title is also inadequate.

Line 242 - do the authors believe that % of FA should be quoted to 1 decimal place??

Line 270 - I disagree that a ruminant lamb is a theoretical model for gaining insight into human conditions. 

Comments on the Quality of English Language

acceptable. The word mutton is misused as it refers to older sheep (Gnerally > 1 year of age). Lamb or growing shepp should be used. 

Round 2

Reviewer 3 Report

Comments and Suggestions for Authors

The Effect of Mulberry Silages Supplementation on carcass fatness and long chain fatty acid composition of growing lambs with traditional corn silage.

by

Cao etal

General Comments

I find that the authors have done little to address the referees comments apart from inserting the details on number of pens and feeding components. I am sceptical of these insertions, when they were totally omitted from the first draft.

These details would have been better entered in a Table format rather the text, so that readers could appreciate the differences, or lack thereof.

I also question the use of supplementary tables in the current draft. MDPI does not appear to have a policy on the use of supplementary tables. I have had a look at Elsevier instructions to authors and I agree with their conclusion that supplementary tables are generally a distraction and are often misused. They are generally not encouraged. Certainly in current paper the authors refer to these tables in the revised text. I would prefer that Supplementary tables 1 and 2 be included in the paper and that their numbering be reversed. It is not logical to have a paper which focuses on subcutaneous fat and intramuscular fat content and yet the data is contained in supplementary tables.

I still have questions that were not addressed by the revised manuscript. Were the animals fed as a restricted or ad libitum diet. What was the energy content of the ration (MJ/kg). If fed ad libitum were the residuals weighed?? What was the average intake of the ration? Was liveweight of the animals recorded???

The design as stated in the redrafted paper is unusual. Why would you have the 5 pens, each of 4 animals, where all animals were blood sampled and then only 1 selected from each pen for slaughter? Was the experimental design comprised by cost constraints?

Given my earlier comments on the experimental unit I am not sure whether the authors have understood the implications of these comments for this statistical analyses. Pen is now the experimental unit (ie 5/treatment, 10 in total) and the between animal variation (with 4 animals per pen) an attempt to reduce sampling error. For slaughter traits there was only 1 animal per pen. For the different blood metabolites and slaughter traits there are only 10 df in the experiment.

Subcutaneous fat and intramuscular fat are the key carcass traits and yet the data is relegated to the supplementary tables. Why???

How is lean meat and bone defined?? What about intermuscular fat??

The authors also ignored my comments on the size of the vertical axes on the graphs. They are TOO SMALL to be read by most readers. They need to be enlarged.

My last point questions the usefulness of the current results. Do the authors believe that feeding mulberry silage to lambs would reduce the saturated (C18:0) and increase unsaturated (C18:1, C18:2, C18:3 etc) FAs intakes to sufficiently cause of a significant change in health status of human patients due to to differences reported in this paper. I doubt it? Give the reader some figures and they can calculate the effect.

I am concerned that lamb fat comprises a large proportion of saturated fatty acids and when lamb is consumed as part of a human diet the differences in this experiment will do little to increase unsaturated fatty acid intake.

My conclusion is that the paper should be rejected.

Specific comments

A better title would be

The effects on carcass fat and long chain fatty acid composition in growing lambs were supplemented with mulberry or corn silage.

In this example the effects on on carcass fatness and FA composition are mentioned 1st with the treatments mentioned 2nd.

Abstract

Line 14 - Lamb meat....

Line 16 - polyunsaturated and saturated fatty acids...

Line 16 - This study used proteomic and metabolic analyses compared a basal ration supplemented with either milage or corn silage

Line 24 - To comply with other scientific papers the authors should use the abrev of C18:3, C18:2 and C18:1 etc throughout the paper

Line 24 - as mentioned in the previous draft this study is not about models to provide guidance for human nutrition. If humans want a low fat diet stop eating lamb??

Line 28 lamb tenderness: intramuscular fat

Introduction

Line 31 - Is there a reference for the generalised statement that lamb can satisfy the higher taste and nutritional values of consumers.

Line 34 - the authors need to be upfront that lamb fat comprises saturated or mono-unsaturated fatty acid. The poly-unsaturated fatty acids only comprises a minor proportion of lamb fat.

Line 27 - to minimise the consumption of saturated fatty acids why not cut lamb from the diet.

Line 39 - Lambs also had a higher content of saturated fatty acids, which makes them harder to metabolise.

Not clear what the authors mean by this sentence?? In terms of fatty acid deposition in the animal saturated fatty acids contain more energy than unsaturated fat fatty acids?. Do they mean harder to metabolise for humans in which I think the phrase is not relevant in this section. Definitely requires a reference if this statement is pursued.

Line 68 why are the authors citing results from immunological studies???. The paper does not mention any immunological results and therefore this section needs deletion.

Line 71 - the pig is a monogastric and the lamb is a ruminant. Is feeding mulberry leaf powder to pigs relevant in this study?? I dispute that it is!!!

Line 78 - subcutaneous carcass fat....

Line 78 - the hypothesis has not improved and need a total rewrite. There are web sites that will take the authors through this.

Material and Methods

Line 84 - there is present and past tense in the text. Make all tense past tense. Given the lack of detail in the last draft I can now comment on the inserted details.

A table is required for feed composition. Was it feed as a restricted or ab libitum ration? were there residuals if so how much. What was the energy density expressed as MJ/kg of dry feed. What was the intake of the lambs? Were the lambs weighed regularly. If so their mean weight (+/- sd) needs to go in table 1. How was the mean (+/- stdev) calculated using pen means, or individual animals?

Line 90 - what does freely fed in the pens mean??

What was the liveweight of the slaughtered lambs?

Line 90 - What was the purpose of analysing the 23 blood metabolites??

Line 97 - what was the definition of subcutaneous fat. Did it include fat beneath the m. cutaneous trunci or only fat exterior to this muscle. Percentage bone and lean are mentioned in the Supplementary tables without a definition of how they were measured.

Line 101 - the statement from the AE Committee needs to go at the start of the M&M

Line 105 - what was the diameter of the LD sample. Where about was the sample taken on the LD?

Line 109 - where did you get the method for shear force. You need a reference?? What were the dimensions of the cooking loss samples. I doubt whether placing the sample in boiling water for 5 minutes would have cooked it sufficiently. Normally sheep meat samples are cooked for half an hour.

Line 112 - Cooking the shear force sample at 18oC for 4 hrs is not a standardised manner of preparing the sample for measuring shear force.

With such a different methodology means that the results cannot be compared and are therefore of questionable value and need to be deleted. How was the sample heated, was it ramped up from 18 to 70oC in the water bath, or were the samples placed directly into a 70oC waterbath????

With the heterogeneous nature of meat the US and Australian shear force procedures generally requires 6 duplicates per sample. Why did the authors only decide 2 replicates. Dimensions of the shear force sample are required?

Line 115 - what protocol did the laboratory follow?? It is not enough to simply say that the samples were processed. Need a citable reference?

Line 171 - I am concerned that there were relatively few animals in the experiment. Nowhere does the text tell me that there were only 10 df for this experiment.

Line 190 - 193 the significance level should be in the figures and tables, not in the stats methods?

Line 198 - Figure 1A) serves no purpose and should be deleted.

Line 202 - The results are a mixture of results from 40 lambs and 10 lambs for slaughter.

I am not surprised the results for slaughter traits were not significant with only 10 df. Did the authors do a power test to determine how many animals were required to pick up significance???

Line 204 - need to define acronyms HDH, LDL CHO etc

Line 211 - are the references to human health relevant to a comparison with lambs. Delete??

Line 216 - I cannot get figures from these figures. Better in a table? Acronyms need to be defined in B)

Line 281 - I find the continual reference to monogastric species (poultry, mice pigs etc) a worry as the experiment was conducted using ruminant lambs which would have changed a large proportion of double bonds (unsaturated) in fatty acids to saturated FAs.

Line 357 - Why would the authors bother including the C18:x notation in Figure 6 and yet not include it in the rest of the paper.

Line 365 - are 2 decimal places required?? I doubt it.

Comments on the Quality of English Language

there are a  number of sentences where eidting is required to clarfy what is meant.
